# Evaluation of Marker Gene-Based In Silico Antimicrobial Resistance Prediction Tools

**DOI:** 10.3390/biology14101405

**Published:** 2025-10-13

**Authors:** Woo Jin Kim, Chorong Hahm, Dongin Kim, Daewon Kim, Ja Young Seo, Jeong Yeal Ahn, Pil Whan Park, Yiel Hea Seo, Joohee Lee

**Affiliations:** 1EONE Laboratories, Incheon 22014, Republic of Korea; oojinkim@eonelab.co.kr (W.J.K.); crhahm@eonelab.co.kr (C.H.);; 2Department of Laboratory Medicine, Gil Medical Center, Gachon University College of Medicine, Incheon 21565, Republic of Korea; jyseojy@gilhospital.com (J.Y.S.); jyahn@gilhospital.com (J.Y.A.); pwpark@gilhospital.com (P.W.P.); seoyh@gilhospital.com (Y.H.S.); music0525@naver.com (J.L.)

**Keywords:** antimicrobial resistance AMR, marker gene-based prediction tools, antimicrobial genes (ARGs), *Escherichia coli*, PICRUSt2, Tax4Fun, MicFunPred

## Abstract

**Simple Summary:**

While 16S rRNA-based predictive functional profiling has proven useful for broad ecological studies, its utility for AMR surveillance remains limited. The results of our study highlight the necessity of integrating specialized AMR databases and improving algorithmic approaches to achieve meaningful accuracy in resistance prediction. These advancements will be essential if marker gene-based tools are to complement or substitute for shotgun metagenomics in the context of clinical or epidemiological AMR monitoring.

**Abstract:**

The monitoring and surveillance of antimicrobial resistance (AMR) is an important procedure in clinical patient management and epidemiological public health. Conventionally, culture-based tools such as disk diffusion methods or broth dilution methods for antibiotic susceptibility tests are used. While culture-independent approaches, such as PICRUSt2, Tax4Fun, or MicFunPred, have recently been tried based on predictive functional profiling using the 16S rRNA marker gene, evaluations of AMR tools are scarce. A total of 20 *E. coli* strains (Carbapenem-resistant (CRE) positive: 10, CRE negative: 10) were used. The AMR phenotype was based on Vitek2 (bioMerieux). DNA was extracted from the 20 strains, and 16S rRNA (V3-V4 region) and shotgun sequencing was carried out. The bioinformatic pipelines were QIIM2 for 16S rRNA and MetaPhlAn4 for shotgun. The functional prediction tools were PICRUSt2, Tax4Fun, and MicFunPred for 16S rRNA and AMRFinderPlus for shotgun. The presence/absence of 23 KEGG numbers regarding AMR in PICRUSt2, Tax4Fun, and MicFunPred were compared to shotgun AMR profiles. The F1 scores were calculated according to each 16S marker gene-based prediction tool using a confusion matrix. A total of 12 classes of antibiotics, including carbapenem, were analyzed. The F1 scores of 16S predictive functional profilers regarding AMR were 0.22 for Tax4Fun, 0.12 for PICRUSt2, and 0.08 for MicFunPred. While Tax4Fun showed the highest F1 score of the three 16S predictive functional profilers, the F1 scores were generally low. Our study highlights the necessity of integrating specialized AMR databases and improving algorithmic approaches to achieve meaningful accuracy in resistance prediction.

## 1. Introduction

Antimicrobial resistance (AMR) is one of the most pressing global health challenges of the 21st century. The escalating rate at which microorganisms develop resistance to antimicrobial agents, coupled with a declining pipeline of new drugs, threatens to return medicine to a pre-antibiotic era where common infections could once again become life-threatening [1,2]. The economic burden of AMR is substantial, leading to increased healthcare costs, prolonged hospital stays, and significant societal losses due to productivity declines [3]. To mitigate this crisis, the robust and continuous surveillance of AMR is paramount. Effective monitoring allows for the timely detection of emerging resistance patterns, informs public health interventions, guides clinical treatment guidelines, and facilitates the development of new diagnostic and therapeutic strategies [4]. Surveillance efforts must extend beyond clinical settings to encompass environmental reservoirs and agricultural practices, recognizing the interconnectedness of human, animal, and environmental health in the context of AMR dissemination [5].

Traditionally, AMR surveillance has relied heavily on culture-based methods, which involve isolating bacterial strains and performing phenotypic susceptibility testing [6]. While invaluable for clinical decision-making, these methods are often labor-intensive, time-consuming, and limited to culturable microorganisms, potentially underestimating the true diversity and prevalence of resistance in complex microbial communities [7]. The advent of molecular techniques, particularly next-generation sequencing (NGS), has revolutionized AMR monitoring by enabling the culture-independent and high-throughput analysis of microbial populations [8]. These methods directly probe the genetic determinants of resistance, offering a more comprehensive and nuanced understanding of the “resistome”—the collection of all antimicrobial resistance genes (ARGs) in a given environment [9]. Molecular approaches include various PCR-based assays for specific ARGs, but more broadly, genome-wide sequencing technologies have gained prominence.

Whole-Metagenome Sequencing (WMS) stands as the most comprehensive and direct method for AMR surveillance. By sequencing the entire genetic material of an organism or a microbial community (metagenomics), WMS allows for the direct identification and quantification of all known and potentially novel ARGs and their genetic context (e.g., location on chromosomes or mobile genetic elements like plasmids) and provides the highest resolution for tracking the spread of resistant clones [10,11]. For carbapenem-resistant *Escherichia coli* (CRE), a critical clinical challenge, WMS is particularly powerful, enabling the precise identification of specific carbapenemase genes (e.g., *bla*_KPC_, *bla*_NDM_, *bla*_OXA-181_), their associated plasmids, and phylogenetic analysis to trace transmission pathways [12,13].

Despite its unparalleled insights, the high cost, extensive bioinformatics expertise, and computational resources required for WMS often limit its widespread application, especially for large-scale epidemiological studies or routine surveillance in resource-constrained settings [5,14]. This practical limitation has prompted the exploration of alternative, more cost-effective approaches that can still provide valuable information for AMR monitoring.

16S rRNA gene sequencing, a widely adopted and more affordable NGS technique, profiles microbial community composition by targeting a conserved bacterial gene with variable regions. While 16S sequencing is excellent for taxonomic identification and understanding shifts in microbial community structure [4,15], it does not directly sequence ARGs. Consequently, its utility for precise AMR monitoring has been debated. The critical challenge lies in inferring functional potential, particularly the presence and abundance of ARGs, solely from taxonomic profiles.

To bridge this gap, several computational tools have been developed, such as PICRUSt2, Tax4Fun, and MicFunPred, which attempt to predict the functional potential of a microbial community based on its 16S rRNA gene-derived taxonomic composition [16,17,18,19]. These tools leverage extensive genomic databases to infer the likely gene content of identified taxa. While such functional prediction tools have shown promise in general ecological studies, their accuracy and reliability for specifically predicting the presence and abundance of critical ARGs, especially in the context of highly mobile resistance elements like those found in CRE, remain a significant area of investigation [20]. The inference relies on the assumption that the functional gene content of detected 16S phylotypes is consistent with the reference genomes in their databases, which may not hold true for all strains, particularly those with extensive horizontal gene transfer of ARGs [21,22].

Given the practical constraints of WGS and the widespread accessibility of 16S rRNA gene sequencing, there is a compelling need to systematically evaluate the usefulness and limitations of 16S sequencing, coupled with functional prediction tools, for monitoring specific AMR patterns. This study aims to investigate the applicability and reliability of 16S rRNA gene sequencing data, analyzed using functional inference tools such as PICRUSt2, Tax4Fun, and MicFunPred, for the detection and monitoring of antimicrobial resistance within *Escherichia coli* CRE colonies. By focusing on *E. coli* CRE, a critical public health concern, we seek to determine the extent to which these indirect methods can accurately reflect the presence and diversity of carbapenem resistance genes, as compared to direct genetic detection methods. The findings will provide crucial insights into the potential utility of 16S-based approaches as a more accessible alternative or complementary tool for AMR surveillance, particularly in scenarios where WGS is not feasible.

## 2. Materials and Methods

### 2.1. Bacterial Strains and Phenotypic Antimicrobial Susceptibility Testing

Ten CRE *Escherichia coli* isolates (P1–P10) and ten carbapenem-susceptible *E. coli* isolates (N1-10) were obtained for this study, isolated and identified from patients hospitalized at general hospitals in Incheon, Republic of Korea. The presence or absence of carbapenem resistance was initially determined using the VITEK^®^ 2 automated microbiology system (bioMérieux, Marcy l’Etoile, France), following the manufacturer’s instructions. In addition to carbapenem resistance, the VITEK^®^ 2 system (AST-N415) was used to determine the antimicrobial susceptibility profiles for a panel of other clinically relevant antibiotics (amoxicillin/clavulanic acid (CA), ampicillin, aztreonam, cefazolin, cefepime, cefotaxime, cefoxitin, ceftazidime, ciprofloxacin, ertapenem, imipenem, piperacillin/tazobactam, amikacin, gentamicin, trimethoprim/sulfamethoxazole, and tigecycline).

### 2.2. DNA Extraction

Genomic DNA was extracted from bacterial colonies that were cultured from a suspension (1 g of feces mixed with 10 mL of PBS) on sheep blood agar at 37 °C for 24 h under aerobic conditions and on gut microbiota media (GMM) and Gifu anaerobic medium agar, modified (GAM) at 35 °C for 48 h in an anaerobic chamber (5% CO_2_). Briefly, colony samples were processed using the QIAamp PowerFecal Pro Kit (Qiagen, Hilden, Germany) according to the manufacturer’s protocol. The extracted DNA was quantified using a Qubit fluorometer (Thermo Fisher Scientific, Waltham, MA, USA) and its quality was assessed using a NanoDrop spectrophotometer (Thermo Fisher Scientific, Waltham, MA, USA). The extracted DNA was stored at −20 °C until further use.

### 2.3. DNA Sequencing

16S rRNA Gene Sequencing [4,15] and Whole-Genome Sequencing (WGS) [11,12,13,14] were performed on the extracted DNA.

#### 2.3.1. 16S rRNA Gene Sequencing

The V3-V4 hypervariable regions of the 16S rRNA gene were amplified from the extracted DNA. Amplicons were sequenced on the Illumina MiSeqDx platform (Illumina, San Diego, CA, USA) using a paired-end sequencing protocol (2 × 300 bp). The sequencing libraries were prepared according to the Illumina 16S Metagenomic Sequencing Library protocols to amplify the V3 and V4 regions. The 4 ng of input gDNA was PCR amplified with 2× KAPA HiFi HotStart ReadyMix buffer (KAPA Biosystems, Wilmington, MA, USA) and 1 μM each of universal F/R PCR primers. The conditions for PCR were as follows: 3 min at 95 °C for heat activation and 25 cycles of 30 s at 95 °C, 30 s at 55 °C, and 30 s at 72 °C, followed by a 5 min final extension at 72 °C. The universal primer pair with Illumina adapter overhang sequences were as follows. V3-F: 5′-TCG TCG GCA GCG TCA GAT GTG TAT AAG AGA CAG CCT ACG GGN GGC WGC AG-3′; V4-R: 5′- GTC TCG TGG GCT CGG AGA TGT GTA TAA GAG ACA GGA CTA CHV GGG TAT CTA ATC C-3′. The PCR product was purified with AMPure beads (Agencourt Bioscience, Beverly, MA, USA). Following purification, 5 μL of the PCR product was PCR-amplified a second time for final library construction containing the index using NexteraXT Indexed Primer (Illumina, San Diego, CA, USA). The cycle condition for the second PCR was the same as that for the first PCR except for eight cycles. The PCR product was purified with AMPure beads. The final purified product was then quantified using the Qubit (Life Technologies, Carlsbad, CA, USA) 2.0 fluorometer following the manufacturer’s instructions and was qualified using the TapeStation D1000 ScreenTape (Agilent Technologies, Waldbronn, Germany). Paired-end (2 × 300 bp) sequencing was performed using the MiSeq™ platform (Illumina, San Diego, CA, USA).

#### 2.3.2. WGS

Genomic libraries were prepared and sequenced on the Illumina MiSeq platform (Illumina, San Diego, CA, USA) using a paired-end sequencing protocol (2 × 150 bp). WGS libraries were prepared from the extracted DNA. Genomic DNA was sheared to an average size of 350 bp using a Qsonica Sonicator (Qsonica, Newtown, CT, USA). The sheared DNA fragments were then used to prepare sequencing libraries with NETXflex Unique Dual Index Bacrcodes (Austin, TX, USA) following the manufacturer’s protocol. Library quality control was performed using a 4200 TapeStation (Agilent Technologies, Santa Clara, CA, USA) for size distribution and a Qubit 3 Fluorometer (Thermo Fisher Scientific, Waltham, MA, USA) for quantification. All reactions were mixed using an MS3 Vortexer (IKA, Staufen, Germany). Final libraries were amplified using a SimpliAmp Thermal Cycler (Thermo Fisher Scientific, Waltham, MA, USA). Pooled libraries were sequenced on an Illumina MiSeq platform (Illumina, San Diego, CA, USA) to generate 2 × 150 bp paired-end reads.

### 2.4. Bioinformatic Analysis

The raw sequencing data from both 16S rRNA gene and WGS were processed using established bioinformatic pipelines, specifically QIIME2 (version 2.2.1), for 16S analysis and MetaPhlAn4 (version 4.0) for WGS analysis, as detailed in the following subsections.

#### 2.4.1. 16S rRNA Gene Sequencing Data Analysis

Initial processing and quality control were performed using the QIIME 2 version 2.2.1 pipeline. This included demultiplexing, quality filtering, denoising with DADA2, and the generation of amplicon sequence variants (ASVs). The functional inference and prediction of AMR genes were performed using several tools. Forward and reverse paired-end 16S rRNA sequences were merged using the tool QIIME 2 (version: 2021.2). The merged sequences were demultiplexed and divided into samples using the barcode sequence of each sample. Using QIIME 2 plugin DADA2, quality control was performed, and the noise was removed, and denoised and filtered amplicon sequencing variants were rarefied to a depth of 4000 sequences per sample.

##### PICRUSt2

Predicted functional profiles of the microbes were generated from the 16S rRNA gene amplicon sequences using PICRUSt2 (version 2.5.2). The analysis pipeline was initiated with the sequence abundance table and the representative sequence file as input. The representative sequences were first placed onto a reference phylogenetic tree using an accelerated maximum parsimony algorithm, with the SEPP method utilized for inserting sequences into the reference phylogeny. Gene family abundances for each ASV were then inferred based on the gene content of nearby reference genomes in the phylogenetic tree. The resulting per-ASV gene family predictions were subsequently combined with the original ASV abundance table to generate a final predicted metagenome for each sample. The predicted genes were functionally annotated based on the KEGG orthologs (KOs) database. All analyses were performed using default parameters as outlined in the PICRUSt2 documentation [17].

##### Tax4Fun

Predicted functional profiles of the microbial communities were generated from the 16S rRNA gene sequences using Tax4Fun (version 0.3.1). The analysis was conducted by first mapping the ASV to the SILVA 123 reference database using BLASTn (version 2.14.1+) to determine the nearest phylogenetic neighbors. Based on this taxonomic information, functional profiles, including KOs, were inferred. The algorithm subsequently normalized the data by accounting for 16S rRNA gene copy numbers to ensure an accurate representation of the functional profile. The final output, which consisted of a table of predicted functional abundances, was then used for downstream statistical analysis [18].

##### MicFunPred

Predicted functional profiles of the microbial communities were generated from the 16S rRNA gene sequences using MICFunPred (version 1.0.0). The pipeline, which is based on the PICRUSt2 workflow, was utilized to infer functional gene abundances. Initially, the representative sequences were phylogenetically placed onto a reference tree using the SEPP algorithm. Subsequently, the gene copy number of KOs was predicted for each sequence based on the nearest sequenced genomes in the reference tree. The per-sequence predictions were then combined with the original abundance table to create a comprehensive functional profile for each sample. The final output, a table of predicted functional abundances, was used for downstream analysis [19].

#### 2.4.2. WGS Data Analysis

The taxonomic composition of the microbial communities was profiled using MetaPhlAn, version 4.0. MetaPhlAn4 uses a set of clade-specific marker genes to accurately estimate the relative abundance of microbial taxa directly from metagenomic sequencing reads. Quality-filtered reads were mapped against the MetaPhlAn4 marker gene database (ChocoPhlAn v2019) using Bowtie2, and taxonomic profiles were generated at multiple levels, from domain to species. Default parameters were used unless otherwise specified [23].

To identify AMR determinants, we employed AMRFinderPlus, developed by the National Center for Biotechnology Information (NCBI). AMRFinderPlus (v3.11.18) detects acquired AMR genes, point mutations associated with resistance, and select virulence factors by aligning input sequences against the curated NCBI AMRFinderPlus database. The tool was run with default parameters, and both protein and translated nucleotide sequence searches were performed to improve sensitivity. Identified AMR genes were classified by drug class, resistance mechanism, and gene family [24].

### 2.5. Statistical Analysis

The performance metrics of antimicrobial resistance prediction were evaluated using precision, recall, and F1 score. Calculations were performed in Microsoft Excel 2016 (Microsoft Corporation, Redmond, WA, USA). Precision was defined as the proportion of true positives among predicted positives, recall as the proportion of true positives among actual positives, and the F1 score as the harmonic mean of precision and recall:F1 = 2 × Precision × Recall/(Precision + Recall)

Manually curated contingency tables were constructed for each prediction tool, and formulas were applied within Excel cells to compute precision, recall, and F1 scores for comparative evaluation.

## 3. Results

### 3.1. Phenotype Characteristics of E. coli Isolates

The antimicrobial susceptibility of 20 *E. coli* isolates was determined using the VITEK^®^ 2 automated microbiology system, with isolates categorized into two groups: P1–P10 and N1–N10. The results, summarized in Table 1, were evaluated based on the Clinical and Laboratory Standards Institute (CLSI) M100-Ed35 [25] guidelines to ensure standardized and reliable interpretation of the data. The isolates showed distinct patterns of AMR, Extended-spectrum beta-lactamase (ESBL) production, and CRE. All ten isolates in the P1–P10 group were positive for ESBL production. This group also exhibited resistance to a wide range of antibiotics, including Amoxicillin/CA, Ampicillin, Aztreonam, Cefazolin, Cefotaxime, Ertapenem, and Piperacillin/tazobactam. Resistance to Cefepime and Imipenem was also common, with some isolates showing intermediate or sensitive phenotypes. Isolates in this group also frequently showed resistance to Ciprofloxacin and Gentamicin. In contrast, all ten isolates in the N1–N10 group were negative for ESBL production. These isolates were generally sensitive to most of the tested antibiotics. However, some isolates within this group displayed resistance to specific agents. For example, N6, N8, N9, and N10 were resistant to Amikacin, and N6, N7, and N8 were resistant to Gentamicin. Isolate N8 was resistant to Ciprofloxacin, and N2 was resistant to Trimethoprim/Sulfamethoxazole. All 20 isolates, both P and N groups, were sensitive to Tigecycline. The collective data indicates that the P1–P10 isolates are multi-drug-resistant strains, while the N1–N10 isolates are generally susceptible, with some exceptions.

### 3.2. Genotype Characteristics of E. coli Isolates

The genotypic characteristics of the 20 *E. coli* isolates were analyzed using WGS by screening for various AMR genes. The presence (P) or absence (N) of these genes in each isolate is detailed in Table 2. The genes were categorized by the class of antibiotic they confer resistance to, including beta-lactams (BL), aminoglycosides (AM), quinolones (QN), lincosamine–macrolide–streptogramin (LMS), macrolide (MA), trimethoprim (TM), tetracycline (TC), sulfonamide (SU), quaternary ammonium (QA), phenicol (PN), lincosamide (LA), and rifamycin (RM). Consistent with the phenotypic data, the P1–P10 isolates harbored a significant number of resistance genes. Specifically, the beta-lactam resistance genes *bla*_KPC-2_ were present in almost all P1–P10 isolates. The ESBL genes *bla*_CTX-M-15_, *bla*_CTX-M-14_, and *bla*_OXA-1_ were also prevalent within this group. Several P-group isolates also carried genes conferring resistance to other antibiotic classes, such as *aac(6′)-lb-cr5* (aminoglycoside-quinolone) and *catB3* (phenicol). The N1–N10 isolates, which were phenotypically susceptible, generally lacked the resistance genes found in the P-group. However, some N-group isolates did harbor specific resistance genes. For example, isolates N6, N7, N8, and N9 were positive for *aac(3)-lld*. Additionally, several N-group isolates contained genes such as *aadA1* (N2), *dfrA1* (N2), *mph(A)* (N2, N4, N6), *sul1* (N2, N4, N6, N7), and *tet(A)* (N2, N4, N6). Overall, the genotypic analysis confirmed that the multi-drug-resistant P-group isolates carry a broad array of resistance genes, particularly those associated with beta-lactam and ESBL resistance. In contrast, the susceptible N-group isolates possess a more limited and distinct set of resistance genes.

### 3.3. Predictive Function Result of PICRUStt2

PICRUSt2 expressed 17 out of 23 KEGG KO IDs related to AMR, while six IDs were not applicable (NA) (Table 3). The expressed KO IDs and their results for the 20 isolates (10 positive, P1–P10; 10 negative, N1–N10) are as follows. Positive (P) in all 20 isolates: the KO IDs for the genes *aph(6)-Ic/Id* (K04343), *strA* (K10673), and *tet(A)* (K08151) were predicted to be present in all 20 isolates by PICRUSt2. Negative (N) in all 20 isolates: the remaining 14 expressed KO IDs, including *bla*_CMY-2_ (K19096), *bla*_SHV_ (K18699), *bla*_KPC_ (K18768), *bla*_TEM_ (K18698), *aadA* (K00984), *aac6-Ib* (K19278), *erm(A/B/C)* (K00561), *mph* (K06979), *dfrA1* (K18589), *sul1* (K18974), *sul2* (K18824), *qacEΔ1* (K18975), *cmlA* (K18552), and *catB* (K00638), were all predicted as negative (N) for all 20 isolates.

There is notable inconsistency between the PICRUSt2 predictions and the WGS results, as PICRUSt2 largely failed to detect the presence of AMR genes that were identified by WGS AMRFINDERplus. While WGS identified at least one positive result for 23 different gene symbols across the 20 isolates, PICRUSt2 predicted the presence of only three KO IDs (*aph(6)-Ic/Id*, *strA*, and *tet(A)*) in most of the isolates. WGS identified multiple positive beta-lactamase genes, including *bla*_OXA-1_ (P1, P3–P9), *bla*_SHV-11_ (P2, P8, P9, P10), *bla*_KPC-2_ (P2–P10), and *bla*_TEM-1_ (P2–P6, P8, P9, N6, N8–N10). In contrast, PICRUSt2 predicted all corresponding KO IDs (*bla*_CMY-2_, *bla*_SHV_, *bla*_KPC_, *bla*_TEM_) to be negative (N) in every single isolate. WGS detected *aadA* in N2 and aadA5 in P2, P5, N6, and N7. PICRUSt2, however, predicted the *aadA* (K00984) KO ID as negative for all isolates. WGS found *tet(A)* to be present in P7–P9 and N2, N4, N6. PICRUSt2 predicted the *tet(A)* (K08151) KO ID as positive for all isolates. This shows a significant over-prediction by PICRUSt2 compared to the WGS results. WGS detected sul1 in P2, P5, N2, N4, N6, and N7. PICRUSt2 predicted the sul1 (K18974) KO ID as negative for all 20 isolates. The same discrepancy was observed for sul2, which WGS found in P8 and N6, while PICRUSt2 predicted it as negative for all isolates.

### 3.4. Predictive Function Result of Tax4Fun

Tax4Fun was able to express 4 out of 23 KEGG KO IDs related to AMR, while the remaining 19 IDs were not expressed, as indicated by “NA” (Table 4). The four expressed KO IDs and their results across the 20 isolates (P1–P10 and N1–N10) are as follows. K00984 (*aadA*): Tax4Fun predicted this gene to be present in all 20 isolates. K10673 (*strA*): This gene was also predicted to be present in all 20 isolates. K08151 (*tet(A)*): Tax4Fun predicted this gene to be present in all 20 isolates. K00638 (*catB*): This gene was predicted to be present in all 20 isolates.

A comparison with the WGS data reveals a significant over-prediction by Tax4Fun for all four expressed KO IDs. While Tax4Fun predicted a positive result for each of these genes in all 20 isolates, WGS data showed a much lower prevalence. K00984 (*aadA*): WGS identified different variations in this gene, including *aadA1*, *aadA2*, and *aadA5*, in a total of six isolates (P2, P5, P8, N2, N6, and N7). Tax4Fun’s prediction of its presence in all 20 isolates represents a significant over-prediction. K10673 (*strA*): The *strA* gene was not detected at all in the WGS results. Tax4Fun’s prediction of its presence in all 20 isolates, therefore, has no corresponding positive WGS result to support it. K08151 (*tet(A)*): WGS detected the *tet(A)* gene in six isolates (P7, P8, P9, N2, N4, and N6) and *tet(B)* in one isolate (P2). Tax4Fun, however, predicted *tet(A)* as positive in all 20 isolates. K00638 (*catB*): WGS found the *catB3* gene in eight isolates (P1, P3–P9). In contrast, Tax4Fun predicted the gene as positive in all 20 isolates. This analysis highlights that Tax4Fun, in this dataset, appears to have a low specificity, consistently predicting the presence of these genes even when WGS, the more direct method, shows they are absent.

### 3.5. Predictive Function Result of MicFunPred

MicFunPred successfully expressed only one out of 23 KEGG KO IDs related to AMR (Table 5). This single expressed KO ID, K00984, corresponds to the gene symbol aadA. For all 20 isolates, MicFunPred predicted this gene to be present, as indicated by “P” (positive). The remaining 22 KO IDs were not expressed by the tool, denoted as “NA” (not applicable). When compared to the WGS data, Micfunpred’s single prediction shows a significant over-prediction. While MicFunPred predicted the *aadA* gene to be present in all 20 isolates, WGS detected different variations in the corresponding gene (*aadA1*, *aadA2*, and *aadA5*) in only six of the isolates. Specifically, the WGS results show the presence of the gene in P2, P5, P8, N2, N6, and N7. This indicates a low specificity for the MicFunPred tool in this particular dataset.

## 4. Discussion

Since the seminal discovery by Woese and Fox [26], the 16S rRNA gene has served as a cornerstone marker for microbial phylogeny and taxonomy, owing to its ubiquity and evolutionary conservation across bacteria and archaea. Building on this foundation, computational approaches have been developed to leverage 16S rRNA gene sequences not only for taxonomic identification but also for functional inference of microbial communities.

The first widely adopted method for this purpose was PICRUSt (Phylogenetic Investigation of Communities by Reconstruction of Unobserved States), introduced by the Huttenhower group in 2013 [16]. PICRUSt employed the phylogenetic placement of 16S rRNA sequences to infer the likely gene content of microbial taxa by mapping to reference genomes, thereby allowing functional predictions at the community level. Subsequently, Tax4Fun, published by Aßhauer et al. [18], presented an alternative nearest-neighbor mapping approach to predict functional profiles from SILVA-based 16S rRNA classifications. PICRUSt was later refined to PICRUSt2 in 2020 [17], with improvements in genome reference coverage, phylogenetic placement algorithms, and expanded functional databases. More recently, MicFunPred was introduced by Dhotre et al. [19] as another predictive framework for functional profiling based on 16S rRNA data.

The central principle underlying these approaches is the inference of KOs, which are then mapped to curated databases such as IMG, GTDB, or SILVA after phylogenetic placement or nearest-neighbor classification. These inferred KO profiles are subsequently used to approximate the functional repertoire of a microbial community, including pathways associated with AMR.

In this study, we specifically evaluated the performance of three marker gene-based functional prediction tools—PICRUSt2, Tax4Fun, and MicFunPred—for their ability to predict AMR-related functions. A total of 20 *E. coli* isolates (10 CRE-positive and 10 CRE-negative) were characterized using Vitek2 for phenotypic resistance profiles, and WGS was performed to establish ground-truth AMR profiles. These shotgun-derived AMR profiles were compared with 16S-based functional predictions from the three tools.

Our results demonstrated that the overall F1 scores were low across all methods, indicating limited predictive capacity of marker gene-based approaches for AMR functions. Of the three tools, Tax4Fun exhibited the highest F1 score (0.22), followed by PICRUSt2 (0.12) and MicFunPred (0.08). However, the difference between Tax4Fun and PICRUSt2 was modest, and importantly, PICRUSt2 yielded the highest number of KEGG KO IDs overall (Table 6). These findings suggest that, while Tax4Fun’s mapping strategy may confer advantages for specific functional categories, the broader genomic coverage of PICRUSt2 enables greater functional breadth.

The low performance of all tools in AMR prediction is likely attributable to three main factors: (i) the mapping algorithms used to link taxonomy to function, (ii) the fidelity of reference databases with respect to AMR genes, and (iii) the inherent resolution limitations of short-read 16S rRNA gene sequencing for resolving species-level differences that correlate with specific resistance genes. Between these, the quality and content of the reference database appear to be the most critical limitation, a problem that is exacerbated by the short-read nature of the majority of the current PICRUSt2 reference genome assemblies. Current general microbial reference databases such as IMG, GTDB, or SILVA are not specifically curated for the complete spectrum of AMR determinants. Consequently, their coverage of resistance-related functions is often incomplete and insufficient for accurate prediction.

To address these limitations, three key strategies can be envisioned. First, the integration of AMR-specific reference databases such as AMRFinderPlus [24] or CARD (Comprehensive Antibiotic Resistance Database) [27] into the predictive frameworks could significantly improve functional annotation fidelity for resistance genes. Furthermore, while laborious and time-consuming, the development of highly curated, custom databases tailored to specific microbial communities or resistance mechanisms could offer unparalleled accuracy and specificity beyond general AMR repositories. Second, improving the input resolution by adopting full-length 16S sequencing is essential to mitigate resolution challenges. Third, advances in machine learning and AI-driven mapping algorithms may enable the more sophisticated inference of function from taxonomic signals, potentially capturing complex associations between phylogeny and resistance traits.

## 5. Conclusions

In conclusion, while 16S rRNA-based predictive functional profiling has proven useful for broad ecological studies, its utility for AMR surveillance remains limited. The results of our study highlight the necessity of integrating specialized AMR databases and improving algorithmic approaches to achieve meaningful accuracy in resistance prediction. These advancements will be essential if marker gene-based tools are to complement or act as substitutes for shotgun metagenomics in the context of clinical or epidemiological AMR monitoring.

## Figures and Tables

**Table 1 biology-14-01405-t001:** Phenotype characteristics of ten carbapenem-resistant *Escherichia coli* (CRE) isolates (P1–P10) and ten carbapenem-susceptible *E. coli* isolates (N1–N10).

ABX	Name	P1	P2	P3	P4	P5	P6	P7	P8	P9	P10	N1	N2	N3	N4	N5	N6	N7	N8	N9	N10
BL	Amoxicillin/CA	R	R	R	R	R	R	R	R	R	R	S	S	S	S	S	S	S	S	S	S
	Ampicillin	R	R	R	R	R	R	R	R	R	R	S	S	S	S	S	S	S	S	S	S
	Aztreonam	R	R	R	R	R	R	R	R	R	R	S	S	S	S	S	S	S	S	S	S
	Cefazolin	R	R	R	R	R	R	R	R	R	R	S	S	S	S	S	S	S	S	S	S
	Cefepime	I	S	I	I	R	R	I	S	R	S	S	S	S	S	S	S	S	S	S	S
	Cefotaxime	R	R	R	R	R	R	R	R	R	R	S	S	S	S	S	S	S	S	S	S
	Cefoxitin	R	S	R	R	I	R	R	I	I	S	S	S	S	S	S	S	S	S	S	S
	Ceftazidime	R	S	R	R	R	R	R	S	R	S	S	S	S	S	S	S	S	S	S	S
	Ciprofloxacin	R	S	R	R	R	R	R	R	R	R	S	S	S	S	S	S	S	R	S	S
	ESBL	P	P	P	P	P	P	P	P	P	P	N	N	N	N	N	N	N	N	N	N
	Ertapenem	R	R	R	R	R	R	R	R	R	R	S	S	S	S	S	S	S	S	S	S
	Imipenem	I	R	R	R	R	R	I	R	R	R	S	S	S	S	S	S	S	S	S	S
	Piperacillin/ tazobactam	R	R	R	R	R	R	R	R	R	R	S	S	S	S	S	S	S	S	S	S
AM	Amikacin	S	S	S	S	S	S	S	S	S	S	S	S	S	S	S	R	S	R	R	R
	Gentamicin	S	S	S	R	S	R	R	S	R	S	S	S	S	S	S	R	R	R	S	S
TM	Trimethoprim/ Sulfamethoxazole	S	R	S	S	R	S	S	R	S	S	S	R	S	S	S	S	S	S	S	S
TC	Tigecycline	S	S	S	S	S	S	S	S	S	S	S	S	S	S	S	S	S	S	S	S

Abbreviations: CA, clavulanic acid; ESBL, extended spectrum beta-lactamases; R, resistant; I, intermediate; S, sensitive; P, positive; N, negative; ABX, antibiotics; BL, beta-lactam; AM, aminoglycoside; TM, trimethoprim; TC, tetracycline.

**Table 2 biology-14-01405-t002:** Genotype characteristics of ten carbapenem-resistant *Escherichia coli* (CRE) isolates (P1–P10) and ten carbapenem-susceptible *E. coli* isolates (N1–N10).

ABX	Gene Symbol	P1	P2	P3	P4	P5	P6	P7	P8	P9	P10	N1	N2	N3	N4	N5	N6	N7	N8	N9	N10
BL	*bla* _OXA_	N	N	N	N	P	N	N	N	N	N	N	N	N	N	N	N	N	N	N	N
	*bla* _OXA-1_	P	N	P	P	P	P	P	P	P	N	N	N	N	N	N	N	N	N	N	N
	*bla* _OXA-181_	P	N	N	N	N	N	N	N	N	N	N	N	N	N	N	N	N	N	N	N
	*bla* _CMY-2_	P	N	N	N	N	N	N	N	N	N	N	N	N	N	N	N	N	N	N	N
	*bla* _SHV-11_	N	P	N	N	N	N	N	P	P	P	N	N	N	N	N	N	N	N	N	N
	*bla* _SHV-12_	N	N	N	N	N	N	P	N	N	N	N	N	N	N	N	N	N	N	N	N
	*bla* _KPC-2_	N	P	P	P	P	P	P	P	P	P	N	N	N	N	N	N	N	N	N	N
	*bla* _CTX-M-27_	N	P	N	N	N	N	N	N	N	N	N	N	N	N	N	N	N	N	N	N
	*bla* _CTX-M-15_	N	N	P	P	P	P	N	N	P	N	N	N	N	N	N	N	N	N	N	N
	*bla* _CTX-M-14_	N	N	P	P	N	P	P	N	P	N	N	N	N	N	N	N	N	N	N	N
	*bla* _CTX-M-65_	N	N	N	N	N	N	N	P	N	N	N	N	N	N	N	N	N	N	N	N
	*bla* _TEM-1_	N	P	P	P	P	P	N	P	N	N	N	N	N	N	N	P	N	P	P	P
AM	*aadA1*	N	N	N	N	N	N	N	N	N	N	N	P	N	N	N	N	N	N	N	N
	*aadA2*	N	N	N	N	N	N	N	P	N	N	N	N	N	N	N	N	N	N	N	N
	*aadA5*	N	P	N	N	P	N	N	N	N	N	N	N	N	N	N	P	P	N	N	N
	*aac(3)-lld*	N	N	N	N	N	N	N	N	N	N	N	N	N	N	N	P	P	P	P	N
	*aac(3)-lle*	N	N	N	P	N	P	P	N	P	N	N	N	N	N	N	N	N	N	N	N
	*aph(3”)-lb*	N	N	N	N	N	N	N	P	N	N	N	N	N	N	N	P	N	N	N	N
	*aph(6)-ld*	N	N	N	N	N	N	N	P	N	N	N	N	N	N	N	P	N	N	N	N
AM-QN	*aac(6′)-lb-cr5*	P	N	P	P	P	P	P	P	P	N	N	N	N	N	N	N	N	N	N	N
LMS	*erm(B)*	N	N	N	N	N	N	N	N	N	N	N	N	N	P	N	N	N	N	N	N
QN	*qnrS1*	P	N	P	P	N	P	N	P	N	N	N	N	N	N	N	N	N	N	N	N
	*qnrS2*	N	N	N	N	N	N	N	P	N	N	N	N	N	N	N	N	N	N	N	N
	*qnrS13*	N	N	N	N	N	N	N	N	N	N	N	N	N	P	N	N	N	N	N	N
MA	*mph(A)*	N	P	N	P	N	P	N	N	N	N	N	P	N	P	N	P	N	N	N	N
TM	*dfrA1*	N	N	N	N	N	N	N	N	N	N	N	P	N	N	N	N	N	N	N	N
	*dfrA5*	N	N	N	N	N	N	N	N	N	N	N	N	N	P	N	N	N	N	N	N
	*dfrA14*	N	N	N	N	N	N	N	P	N	N	N	N	N	N	N	N	N	N	N	N
	*dfrA17*	N	P	N	N	P	N	N	N	N	N	N	N	N	N	N	P	P	N	N	N
TC	*tet(A)*	N	N	N	N	N	N	P	P	P	N	N	P	N	P	N	P	N	N	N	N
	*tet(B)*	N	P	N	N	N	N	N	N	N	N	N	N	N	N	N	N	N	N	N	N
SU	*sul1*	N	P	N	N	P	N	N	N	N	N	N	P	N	P	N	P	P	N	N	N
	*sul2*	N	N	N	N	N	N	N	P	N	N	N	N	N	N	N	P	N	N	N	N
QA	*qacE∆1*	N	P	N	N	P	N	N	P	N	N	N	P	N	P	N	P	P	N	N	N
PN	*catB3*	P	N	P	P	P	P	P	P	P	N	N	N	N	N	N	N	N	N	N	N
	*floR*	N	N	N	N	N	N	N	P	N	N	N	N	N	N	N	N	N	N	N	N
LA	*lnu(F)*	N	N	N	N	N	N	N	P	N	N	N	N	N	N	N	N	N	N	N	N
RM	*arr-3*	N	N	N	N	N	N	N	P	N	N	N	N	N	N	N	N	N	N	N	N

Abbreviations: BL, beta-lactam; AM, aminoglycoside; QN, quinolone; LMS, lincosamine–macrolide–streptogramin; MA, macrolide; TM, trimethoprim; TC, tetracycline; SU, sulfonamide; QA, quaternary ammonium; PN, phenicol; LA, lincosamide; RM, rifamycin; P, positive; N, negative.

**Table 3 biology-14-01405-t003:** Predictive function result of PICRUSt2 for ten carbapenem-resistant *Escherichia coli* (CRE) isolates (P1–P10) and ten carbapenem-susceptible *E. coli* isolates (N1–N10).

ABX	KO ID	Gene Symbol	P1	P2	P3	P4	P5	P6	P7	P8	P9	P10	N1	N2	N3	N4	N5	N6	N7	N8	N9	N10
BL	K18790	*bla* _OXA-1_	NA	NA	NA	NA	NA	NA	NA	NA	NA	NA	NA	NA	NA	NA	NA	NA	NA	NA	NA	NA
	K18976	*bla* _OXA-48_	NA	NA	NA	NA	NA	NA	NA	NA	NA	NA	NA	NA	NA	NA	NA	NA	NA	NA	NA	NA
	K19096	*bla* _CMY-2_	N	N	N	N	N	N	N	N	N	N	N	N	N	N	N	N	N	N	N	N
	K18699	*bla* _SHV_	N	N	N	N	N	N	N	N	N	N	N	N	N	N	N	N	N	N	N	N
	K18768	bla_KPC_	N	N	N	N	N	N	N	N	N	N	N	N	N	N	N	N	N	N	N	N
	K18767	bla_CTX-M_	NA	NA	NA	NA	NA	NA	NA	NA	NA	NA	NA	NA	NA	NA	NA	NA	NA	NA	NA	NA
	K18698	bla_TEM_	N	N	N	N	N	N	N	N	N	N	N	N	N	N	N	N	N	N	N	N
AM	K00984	*aadA*	N	N	N	N	N	N	N	N	N	N	N	N	N	N	N	N	N	N	N	N
	K19275	*aac(3)-ll*	NA	NA	NA	NA	NA	NA	NA	NA	NA	NA	NA	NA	NA	NA	NA	NA	NA	NA	NA	NA
	K19278	*aac6-Ib*	N	N	N	N	N	N	N	N	N	N	N	N	N	N	N	N	N	N	N	N
	K04343	*aph(6)-Ic/Id*	P	P	P	P	P	P	P	P	P	P	P	P	P	P	P	P	P	P	P	P
	K10673	*strA*	P	P	P	P	P	P	P	P	P	P	P	P	P	P	P	P	P	P	P	P
LMS	K00561	*erm(A/B/C)*	N	N	N	N	N	N	N	N	N	N	N	N	N	N	N	N	N	N	N	N
MA	K06979	*mph*	N	N	N	N	N	N	N	N	N	N	N	N	N	N	N	N	N	N	N	N
TM	K18589	*dfrA1*	N	N	N	N	N	N	N	N	N	N	N	N	N	N	N	N	N	N	N	N
TC	K08151	*tet(A)*	P	P	P	P	P	P	P	P	P	P	P	P	P	P	P	P	P	P	P	P
SU	K18974	*sul1*	N	N	N	N	N	N	N	N	N	N	N	N	N	N	N	N	N	N	N	N
	K18824	*sul2*	N	N	N	N	N	N	N	N	N	N	N	N	N	N	N	N	N	N	N	N
QA	K18975	*qacE* *Δ1*	N	N	N	N	N	N	N	N	N	N	N	N	N	N	N	N	N	N	N	N
QN	K18552	*cmlA*	N	N	N	N	N	N	N	N	N	N	N	N	N	N	N	N	N	N	N	N
PN	K00638	*catB*	N	N	N	N	N	N	N	N	N	N	N	N	N	N	N	N	N	N	N	N
LA	K18236	*lnuB_F*	NA	NA	NA	NA	NA	NA	NA	NA	NA	NA	NA	NA	NA	NA	NA	NA	NA	NA	NA	NA
RM	K21288	*arr-2*	NA	NA	NA	NA	NA	NA	NA	NA	NA	NA	NA	NA	NA	NA	NA	NA	NA	NA	NA	NA

Abbreviations: BL, beta-lactam; AM, aminoglycoside; QN, quinolone; LMS, lincosamine–macrolide–streptogramin; MA, macrolide; TM, trimethoprim; TC, tetracycline; SU, sulfonamide; QA, quaternary ammonium; PN, phenicol; LA, lincosamide; RM, rifamycin; P, positive; N, negative; NA, not applicable.

**Table 4 biology-14-01405-t004:** Predictive function result of Tax4Fun for ten carbapenem-resistant *Escherichia coli* (CRE) isolates (P1–P10) and ten carbapenem-susceptible *E. coli* isolates (N1–N10).

ABX	KO ID	Gene Symbol	P1	P2	P3	P4	P5	P6	P7	P8	P9	P10	N1	N2	N3	N4	N5	N6	N7	N8	N9	N10
BL	K18790	*bla* _OXA-1_	NA	NA	NA	NA	NA	NA	NA	NA	NA	NA	NA	NA	NA	NA	NA	NA	NA	NA	NA	NA
	K18976	*bla* _OXA-48_	NA	NA	NA	NA	NA	NA	NA	NA	NA	NA	NA	NA	NA	NA	NA	NA	NA	NA	NA	NA
	K19096	*bla* _CMY-2_	NA	NA	NA	NA	NA	NA	NA	NA	NA	NA	NA	NA	NA	NA	NA	NA	NA	NA	NA	NA
	K18699	*bla* _SHV_	NA	NA	NA	NA	NA	NA	NA	NA	NA	NA	NA	NA	NA	NA	NA	NA	NA	NA	NA	NA
	K18768	bla_KPC_	NA	NA	NA	NA	NA	NA	NA	NA	NA	NA	NA	NA	NA	NA	NA	NA	NA	NA	NA	NA
	K18767	bla_CTX-M_	NA	NA	NA	NA	NA	NA	NA	NA	NA	NA	NA	NA	NA	NA	NA	NA	NA	NA	NA	NA
	K18698	bla_TEM_	NA	NA	NA	NA	NA	NA	NA	NA	NA	NA	NA	NA	NA	NA	NA	NA	NA	NA	NA	NA
AM	K00984	*aadA*	P	P	P	P	P	P	P	P	P	P	P	P	P	P	P	P	P	P	P	P
	K19275	*aac(3)-ll*	NA	NA	NA	NA	NA	NA	NA	NA	NA	NA	NA	NA	NA	NA	NA	NA	NA	NA	NA	NA
	K19278	*aac6-Ib*	NA	NA	NA	NA	NA	NA	NA	NA	NA	NA	NA	NA	NA	NA	NA	NA	NA	NA	NA	NA
	K04343	*aph(6)-Ic/Id*	NA	NA	NA	NA	NA	NA	NA	NA	NA	NA	NA	NA	NA	NA	NA	NA	NA	NA	NA	NA
	K10673	*strA*	P	P	P	P	P	P	P	P	P	P	P	P	P	P	P	P	P	P	P	P
LMS	K00561	*erm(A/B/C)*	NA	NA	NA	NA	NA	NA	NA	NA	NA	NA	NA	NA	NA	NA	NA	NA	NA	NA	NA	NA
MA	K06979	*mph*	NA	NA	NA	NA	NA	NA	NA	NA	NA	NA	NA	NA	NA	NA	NA	NA	NA	NA	NA	NA
TM	K18589	*dfrA1*	NA	NA	NA	NA	NA	NA	NA	NA	NA	NA	NA	NA	NA	NA	NA	NA	NA	NA	NA	NA
TC	K08151	*tet(A)*	P	P	P	P	P	P	P	P	P	P	P	P	P	P	P	P	P	P	P	P
SU	K18974	*sul1*	NA	NA	NA	NA	NA	NA	NA	NA	NA	NA	NA	NA	NA	NA	NA	NA	NA	NA	NA	NA
	K18824	*sul2*	NA	NA	NA	NA	NA	NA	NA	NA	NA	NA	NA	NA	NA	NA	NA	NA	NA	NA	NA	NA
QA	K18975	*qacE* *Δ1*	NA	NA	NA	NA	NA	NA	NA	NA	NA	NA	NA	NA	NA	NA	NA	NA	NA	NA	NA	NA
QN	K18552	*cmlA*	NA	NA	NA	NA	NA	NA	NA	NA	NA	NA	NA	NA	NA	NA	NA	NA	NA	NA	NA	NA
PN	K00638	*catB*	P	P	P	P	P	P	P	P	P	P	P	P	P	P	P	P	P	P	P	P
LA	K18236	*lnuB_F*	NA	NA	NA	NA	NA	NA	NA	NA	NA	NA	NA	NA	NA	NA	NA	NA	NA	NA	NA	NA
RM	K21288	*arr-2*	NA	NA	NA	NA	NA	NA	NA	NA	NA	NA	NA	NA	NA	NA	NA	NA	NA	NA	NA	NA

Abbreviations: BL, beta-lactam; AM, aminoglycoside; QN, quinolone; LMS, lincosamine–macrolide–streptogramin; MA, macrolide; TM, trimethoprim; TC, tetracycline; SU, sulfonamide; QA, quaternary ammonium; PN, phenicol; LA, lincosamide; RM, rifamycin; P, positive; NA, not applicable.

**Table 5 biology-14-01405-t005:** Predictive function result of MicFunPred for ten carbapenem-resistant *Escherichia coli* (CRE) isolates (P1–P10) and ten carbapenem-susceptible *E. coli* isolates (N1–N10).

ABX	KO ID	Gene Symbol	P1	P2	P3	P4	P5	P6	P7	P8	P9	P10	N1	N2	N3	N4	N5	N6	N7	N8	N9	N10
BL	K18790	*bla* _OXA-1_	NA	NA	NA	NA	NA	NA	NA	NA	NA	NA	NA	NA	NA	NA	NA	NA	NA	NA	NA	NA
	K18976	*bla* _OXA-48_	NA	NA	NA	NA	NA	NA	NA	NA	NA	NA	NA	NA	NA	NA	NA	NA	NA	NA	NA	NA
	K19096	*bla* _CMY-2_	NA	NA	NA	NA	NA	NA	NA	NA	NA	NA	NA	NA	NA	NA	NA	NA	NA	NA	NA	NA
	K18699	*bla* _SHV_	NA	NA	NA	NA	NA	NA	NA	NA	NA	NA	NA	NA	NA	NA	NA	NA	NA	NA	NA	NA
	K18768	bla_KPC_	NA	NA	NA	NA	NA	NA	NA	NA	NA	NA	NA	NA	NA	NA	NA	NA	NA	NA	NA	NA
	K18767	bla_CTX-M_	NA	NA	NA	NA	NA	NA	NA	NA	NA	NA	NA	NA	NA	NA	NA	NA	NA	NA	NA	NA
	K18698	bla_TEM_	NA	NA	NA	NA	NA	NA	NA	NA	NA	NA	NA	NA	NA	NA	NA	NA	NA	NA	NA	NA
AM	K00984	*aadA*	P	P	P	P	P	P	P	P	P	P	P	P	P	P	P	P	P	P	P	P
	K19275	*aac(3)-ll*	NA	NA	NA	NA	NA	NA	NA	NA	NA	NA	NA	NA	NA	NA	NA	NA	NA	NA	NA	NA
	K19278	*aac6-Ib*	NA	NA	NA	NA	NA	NA	NA	NA	NA	NA	NA	NA	NA	NA	NA	NA	NA	NA	NA	NA
	K04343	*aph(6)-Ic/Id*	NA	NA	NA	NA	NA	NA	NA	NA	NA	NA	NA	NA	NA	NA	NA	NA	NA	NA	NA	NA
	K10673	*strA*	NA	NA	NA	NA	NA	NA	NA	NA	NA	NA	NA	NA	NA	NA	NA	NA	NA	NA	NA	NA
LMS	K00561	*erm(A/B/C)*	NA	NA	NA	NA	NA	NA	NA	NA	NA	NA	NA	NA	NA	NA	NA	NA	NA	NA	NA	NA
MA	K06979	*mph*	NA	NA	NA	NA	NA	NA	NA	NA	NA	NA	NA	NA	NA	NA	NA	NA	NA	NA	NA	NA
TM	K18589	*dfrA1*	NA	NA	NA	NA	NA	NA	NA	NA	NA	NA	NA	NA	NA	NA	NA	NA	NA	NA	NA	NA
TC	K08151	*tet(A)*	NA	NA	NA	NA	NA	NA	NA	NA	NA	NA	NA	NA	NA	NA	NA	NA	NA	NA	NA	NA
SU	K18974	*sul1*	NA	NA	NA	NA	NA	NA	NA	NA	NA	NA	NA	NA	NA	NA	NA	NA	NA	NA	NA	NA
	K18824	*sul2*	NA	NA	NA	NA	NA	NA	NA	NA	NA	NA	NA	NA	NA	NA	NA	NA	NA	NA	NA	NA
QA	K18975	*qacE* *Δ* *1*	NA	NA	NA	NA	NA	NA	NA	NA	NA	NA	NA	NA	NA	NA	NA	NA	NA	NA	NA	NA
QN	K18552	*cmlA*	NA	NA	NA	NA	NA	NA	NA	NA	NA	NA	NA	NA	NA	NA	NA	NA	NA	NA	NA	NA
PN	K00638	*catB*	NA	NA	NA	NA	NA	NA	NA	NA	NA	NA	NA	NA	NA	NA	NA	NA	NA	NA	NA	NA
LA	K18236	*lnuB_F*	NA	NA	NA	NA	NA	NA	NA	NA	NA	NA	NA	NA	NA	NA	NA	NA	NA	NA	NA	NA
RM	K21288	*arr-2*	NA	NA	NA	NA	NA	NA	NA	NA	NA	NA	NA	NA	NA	NA	NA	NA	NA	NA	NA	NA

Abbreviations: BL, beta-lactam; AM, aminoglycoside; QN, quinolone; LMS, lincosamine–macrolide–streptogramin; MA, macrolide; TM, trimethoprim; TC, tetracycline; SU, sulfonamide; QA, quaternary ammonium; PN, phenicol; LA, lincosamide; RM, rifamycin; P, positive; NA, not applicable.

**Table 6 biology-14-01405-t006:** Comparative features of marker gene-based predictive function tools.

Feature	PICRUSt2	Tax4Fun	MicFunPred
F1 score in this study	0.12	0.22	0.08
Expressed KO IDs in this study	17/23	4/23	1/23
Version	2.5.2	0.3.1	1.0.0
Prediction Method	Phylogenetic placement and hidden-state prediction.	Nearest-neighbor mapping based on sequence similarity.	Not a well-established or commonly cited tool in this field.
Core Database	A large, curated database of reference genomes (e.g., Integrated Microbial Genomes, GTDB).	SILVA database.	Greengnes, SILVA, EZBiocloud
Input Type	Amplicon Sequence Variants (ASVs) or OTUs. Works with any denoising algorithm.	OTUs assigned to the SILVA database.	Amplicon Sequence Variants (ASVs) or OTUs.
Accuracy	Generally considered highly accurate, especially with its expanded database and phylogenetic approach.	Provides a good approximation; some studies have found good correlations with shotgun data.	N/A
Flexibility	Highly flexible. Can incorporate custom reference databases.	Primarily linked to the SILVA database.	N/A
Popularity	Very popular and widely used in the field.	Well-established and has a significant user base.	Not a mainstream or well-documented tool for this purpose.
Country	USA	Germany	India
References	[17]	[18]	[19]

## Data Availability

Data is contained within the article.

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
