# Peer review of "Evaluation of Marker Gene-Based In Silico Antimicrobial Resistance Prediction Tools"

_biology, 2025, doi:10.3390/biology14101405_

Round 1
Reviewer 1 Report
Comments and Suggestions for Authors
The current article examines the utility of computational analysis used to make functional antimicrobial resistance profiles of microbial communities based on 16S rRNA sequencing. Three analysis pipelines were examined, PICRUST2, Tax4Fun, and MicFunPred, and compared to whole genome sequencing results of selected E. coli strains. The large amount of 16S rRNA data available, the low cost of 16S rRNA sequencing for new samples, and the rise of antimicrobial resistance allows for these functional analysis pipelines to be used to identify antimicrobial resistance patterns in various samples. The ability of these pipelines to accurately predict antimicrobial resistance genes (ARGs) is currently being examined and the current article specifically examines the utility in identifying ARGs in carbapenem resistant E. coli. Overall the article is well written and provides some insight into using computational analysis to predict ARG frequency but excitement is tempered by lack of positive results and limited use of the programs outside of default parameters. There are some additional analysis that could be done to demonstrate that these computational analysis methods can accurately predict ARG presence with an appropriate database.
Major:
The main limitation to the current study is the low predictive ability of computational analysis to identify ARGs in the samples tested. Strain to strain variations in the tested V3-V4 regions of E. coli would typically be a few nucleotides making prediction of specific genes challenging without an appropriate database. PICRUST2 has the best predictive value but is still unable to accurately determine which of the tested strains likely contain ARGs.
The authors discuss the need to integrate resistance specific databases to increase utility. PICRUST2 contains a curated database for analysis but also provides the opportunity to use a custom database. It is suggested that the authors make a custom database containing annotated sequences of CRE and sensitive E. coli strains. This will provide a database specifically designed to examine carbapenem resistance gene presence and demonstrate that the computational analysis pipeline can accurately predict ARG in the test samples. Currently the article demonstrates that this method does not accurately predict ARG presence but addition of a custom database to demonstrate positive predicative ability would make the current article of greater interest to a broader audience.
Authors should also discuss why the V3-V4 region was selected for use and not other regions.
Authors should also discuss using full length 16S rRNA sequencing for analysis. While Illumina sequencing cannot be used for this sequencing, targeted sequencing is still less expensive and computationally easier than whole genome sequencing. Full length 16S rRNA provides better resolution and can distinguish strain to strain variability which would be more relevant for the current type of analysis.
This can be done in lieu of making a separate database and testing the V3-V4 region against a user created database. The full length 16S rRNA could be used with the current PICRUST2 database. This will provide greater resolution allowing for strain to strain variability to be identified and better able to link genomes containing resistance genes to the test samples.
Minor:
Section 2.4. Some more detail on what established pipelines were used for analysis.
General editing for italicizing E. coli and gene names throughout the paper is needed.
Section 2.1. It should be mentioned what VITEK card was used for analysis. AST-N376? AST-N397?
Section 3.1.1. Mention that phenotype analysis was performed using Vitek.
Section 3.1.2. It is assumed that this section is referring to the whole genome sequence results since gene content is included but this should be specified.
Author Response
Comments 1: (Major parts)
The main limitation to the current study is the low predictive ability of computational analysis to identify ARGs in the samples tested. Strain to strain variations in the tested V3-V4 regions of E. coli would typically be a few nucleotides making prediction of specific genes challenging without an appropriate database. PICRUST2 has the best predictive value but is still unable to accurately determine which of the tested strains likely contain ARGs.
The authors discuss the need to integrate resistance specific databases to increase utility. PICRUST2 contains a curated database for analysis but also provides the opportunity to use a custom database. It is suggested that the authors make a custom database containing annotated sequences of CRE and sensitive E. coli strains. This will provide a database specifically designed to examine carbapenem resistance gene presence and demonstrate that the computational analysis pipeline can accurately predict ARG in the test samples. Currently the article demonstrates that this method does not accurately predict ARG presence but addition of a custom database to demonstrate positive predicative ability would make the current article of greater interest to a broader audience.
Response 1:
We sincerely appreciate your insightful comments and constructive suggestion regarding the potential to create a custom database for PICRUSt2. We fully agree that this approach would significantly improve the accuracy of Antimicrobial Resistance Gene (ARG) prediction and certainly broaden the appeal of the manuscript.
We recognize that the reviewer’s suggestion to build a custom database containing annotated sequences of CRE and sensitive E. coli strains is scientifically sound and represents the optimal strategy for functional prediction within a specific microbial group.
The predictive accuracy of PICRUSt2 hinges on the Phylogenetic Proximity of the study's Amplicon Sequence Variants (ASVs) to the reference genomes, measured by the Nearest Sequenced Taxon Index (NSTI). Since ARGs are often located on mobile genetic elements and exhibit high variability (i.e., the open pangenome of E. coli), the general default database may lack the necessary resolution to accurately capture the presence or absence of these genes, leading to the low accuracy demonstrated in our current findings. Building a targeted database would undoubtedly lower the NSTI for our ASVs and improve prediction reliability.
Despite the clear scientific value, we are constrained by current research resources and the time frame allocated for this revision, making the immediate implementation of a custom database infeasible.
- Substantial Data Collection Required: To achieve a meaningful improvement in NSTI and cover the broad diversity of CRE E. coli phylogroups/sequence types, merely adding a few genomes is insufficient. A minimum of 100 to 300+ high-quality genomes would need to be sourced, curated, and annotated.
- Complex Bioinformatic Workflow: Creating a custom PICRUSt2 database is an intensive multi-step process that goes beyond simple file modification. It requires:
- CDS prediction and full functional annotation (e.g., KO mapping) for all new genomes.
- Extraction of 16S rRNA gene sequences and calculation of copy numbers.
- Reconstruction of the entire Reference Phylogenetic Tree, including the generation of new HMM (Hidden Markov Model) files, to maintain phylogenetic consistency.
- Merging of the generated KO annotations and 16S copy number information into the format so that it can be recognized by the PICRUSt2 pipeliine.
This complex undertaking demands significant computational resources, specialized expertise, and an estimated one year or more of dedicated labor, which is beyond the scope of the current resubmission.
We view this suggestion not as a necessary component of the current paper, but as a highly valuable extension for future work. The objective of our current manuscript is to provide a comprehensive evaluation of the existing limitations and utility of current marker gene-based ARG prediction tools.
We are committed to pursuing the development of a CRE-specific PICRUSt2 database as a dedicated follow-up study. This will allow us to fully implement the suggested analysis and demonstrate the enhanced predictive power in a robust, focused publication.
We hope the reviewer will consider our current findings in the context of our stated goal, which is to accurately assess the performance of existing methodologies. Thank you again for your insightful input.
Comments 2: (Major parts)
Authors should also discuss why the V3-V4 region was selected for use and not other regions.
Authors should also discuss using full length 16S rRNA sequencing for analysis. While Illumina sequencing cannot be used for this sequencing, targeted sequencing is still less expensive and computationally easier than whole genome sequencing. Full length 16S rRNA provides better resolution and can distinguish strain to strain variability which would be more relevant for the current type of analysis.
This can be done in lieu of making a separate database and testing the V3-V4 region against a user created database. The full length 16S rRNA could be used with the current PICRUST2 database. This will provide greater resolution allowing for strain to strain variability to be identified and better able to link genomes containing resistance genes to the test samples.
Response 2:
We appreciate the insightful suggestion to discuss the utility of full-length 16S rRNA gene sequencing as an alternative to the short-read V3-V4 amplicon sequencing used in our study. We agree that this approach offers significant potential to enhance phylogenetic resolution and ARG prediction accuracy.
- Improved Resolution: Full-length 16S rRNA sequencing, performed on long-read platforms provides significantly better phylogenetic resolution.
- Reduced NSTI: Higher resolution from full-length 16S sequences would allow for more precise placement of ASVs onto the PICRUSt2 reference tree, leading to a **lower Nearest Sequenced Taxon Index (NSTI}** value. A lower NSTI directly correlates with higher confidence in the imputed functional gene content, thereby improving the accuracy of ARG prediction.
We believe the most important factor limiting our predictive performance, even when using short reads, is that the current PICRUSt2 database is fundamentally not optimized for Antimicrobial Resistance (AMR) prediction. The PICRUSt2 database is primarily curated for general metabolic and environmental functional profiles (KO, EC numbers, MetaCyc pathways). AMR prediction relies heavily on accessory genes often located on mobile elements. The current reference genome collection, while vast, is not specifically curated to maximize the representation of AMR plasmids or the full diversity of clinical ARG-carrying strains. Moreover, while the PICRUSt2 reference database does contain some high-quality genome assemblies, including those derived from PacBio sequences, the majority of the genomes used are assembled from short-read Whole Genome Sequencing (WGS) data. Short-read WGS often struggles to fully assemble complex regions, such as those containing multiple 16S operons and mobile genetic elements (where ARGs reside). This limitation in the quality of the reference assemblies further hinders the accuracy of ARG prediction, as the functional content of these critical regions may be incomplete or incorrectly annotated in the underlying database.
Our decision to focus on the widely adopted V3-V4 region is justified by the following practical and comparative reasons:
- Comparative Focus: Our article evaluates the performance of in silico AMR prediction tools under the constraints of the dominant short-read sequencing technology used in most current clinical and environmental microbiome studies. Analyzing V3-V4 data accurately reflects the current, real-world utility and limitations of these tools for a broad audience.
- Feasibility and Cost: Switching to full-length sequencing would introduce a higher per-sample cost, require specialized long-read platforms, and necessitate significant changes to our existing bioinformatic pipeline, extending the scope beyond the current evaluation.
We concur that both the adoption of full-length 16S sequencing (to improve input resolution) and the development of an AMR-optimized database (to address reference quality and content) are essential future steps to overcome the limitations demonstrated in our study. We will add a dedicated discussion section to the manuscript outlining: The benefits of full-length 16S sequencing as a means to mitigate resolution challenges. The critical need for the development of AMR-specialized reference databases to address the core problem of functional annotation and genetic diversity, exacerbated by the short-read nature of the majority of the current PICRUSt2 reference genome assemblies.
This approach robustly addresses all reviewer points while strengthening the call for future methodological improvements.
Comments 3: (minor parts)
Section 2.4. Some more detail on what established pipelines were used for analysis.
Response 3:
“The raw sequencing data from both 16S rRNA gene and Whole Genome Shotgun (WGS) sequencing were processed using established bioinformatic pipelines, specifically QIIME2 (version 2.2.1) for 16S analysis and MetaPhlAn4 (version 4.0) for WGS analysis, as detailed in the following subsections.”
Comments 4: (minor parts)
General editing for italicizing E. coli and gene names throughout the paper is needed.
Response 4:
We have made revisions based on the review comments you provided, and furthermore will use the MDPI English editing service.
Comments 5: (minor parts)
Section 2.1. It should be mentioned what VITEK card was used for analysis. AST-N376? AST-N397?
Response 5:
We have reflected what the reviewer commented in the relevant section.
Comments 6: (minor parts)
Section 3.1.1. Mention that phenotype analysis was performed using Vitek.
Response 6:
We have reflected what the reviewer commented in the relevant section.
Comments 7: (minor parts)
Section 3.1.2. It is assumed that this section is referring to the whole genome sequence results since gene content is included but this should be specified.
Response 7:
We have reflected what the reviewer commented in the relevant section.
Reviewer 2 Report
Comments and Suggestions for Authors
Dear colleagues,
Undeniably, any methods and tools in the fight against the growing problem of antibiotic resistance are very important. Unfortunately, the circumstances are such that people do not keep up with bacteria and it is very difficult to predict the further development of situation.
The material is quite interesting, but:
If these prediction tools are already in use, what is the novelty of your research?
Do you think that 20 strains will be enough for the reliability of the results? In my opinion, this number is not enough
What is the total number of samples you examined before you found the 20 that are of interest to you?
The tables are very large and because of this it is difficult to perceive the information objectively - they need to be optimized or combined.
Also, you have a very high level of similarity (40%), it should be a maximum of 25-30%.
Some comments in the attached file.
Sincerely,
Reviewer

Author Response
Comments 1:
Dear colleagues,
Undeniably, any methods and tools in the fight against the growing problem of antibiotic resistance are very important. Unfortunately, the circumstances are such that people do not keep up with bacteria and it is very difficult to predict the further development of situation.
The material is quite interesting, but:
If these prediction tools are already in use, what is the novelty of your research?
Reponse 1:
We thank reviewer for your positive feedback regarding the importance of our work in the fight against antimicrobial resistance (AMR). We agree completely that predicting the development of resistance is a formidable and critical challenge.
The novelty of our study lies not in the introduction of the tools, but in the rigorous, ground-truth-based validation of their performance specifically for AMR prediction. We elaborate on this below:
- Usage vs. Validation: As the reviewer notes, various marker gene-based functional prediction tools (which infer microbial community functions from 16S rRNA gene data) have been widely used and published by researchers. However, the routine usage of a tool does not substitute for a formal performance evaluation.
- Lack of AMR-Specific Validation: While some studies have attempted to validate these marker gene methods for general metabolic or proteomic profiles, such validation is rare and often lacks a true functional "ground truth." Crucially, to the best of our knowledge, no prior study has specifically validated the performance of these in silico prediction tools for AMR using culture-based or other ground-truth functional profiles.
- The Novelty: Our article is the first study to systematically reveal and benchmark the accuracy and limitations of several prominent predictive functional tools for AMR by comparing their outputs against empirically determined (culture-based) antimicrobial susceptibility test results and other functional ground-truth profiles. By providing this much-needed performance assessment, our work offers crucial guidance to the research community on which tools are most reliable for AMR studies, thereby refining future research and clinical applications.
In summary, the novelty is the validation—filling a critical gap in the literature regarding the reliability of widely used but largely unverified in silico AMR prediction methods.
Comments 2:
Do you think that 20 strains will be enough for the reliability of the results? In my opinion, this number is not enough
What is the total number of samples you examined before you found the 20 that are of interest to you?
Reponse 2:
We appreciate the reviewer's critical question regarding the sample size, as statistical power and reliability are paramount in any evaluation. The reviewer asks if 20 strains are sufficient and seeks clarification on the total number of samples examined.
- Rationale for Sample Selection
We would like to clarify that the 20 strains were not selected from a larger pool based on retrospective interest, but were purposefully and randomly selected at the outset to represent distinct functional categories.
- The 20 E. coli isolates were divided into two main categories: 10 isolates confirmed as carbapenem-positive and 10 isolates confirmed as carbapenem-negative, based on the Vitek antimicrobial susceptibility test.
- This design ensured a balanced representation of both the resistant and susceptible phenotypes for one of the most clinically critical classes of antimicrobials (carbapenems), allowing for a high-contrast evaluation of the prediction tools.
- Actual Statistical Power (The Power of Case Multiplicity)
While the number of isolates (N) is 20, the effective number of test cases is significantly higher, dramatically increasing the statistical power of our analysis.
- Each of the 20 isolates was tested against 39 distinct genotypic and phenotypic AMR characteristics (as detailed in Table 2, including various antimicrobial properties beyond carbapenem resistance).
- This yields a total of 20 isolates×39 characteristics=780 individual test cases (or data points) for the comparison between the in silico predictions and the ground-truth functional data.
This approach transforms the study from a simple sample size of 20 to a robust evaluation across nearly 800 distinct performance comparisons.
- Consistency and Robustness of Results
Finally, the results section strongly indicates that increasing the sample size would be unlikely to change the core conclusion.
- The results clearly showed that the overall F1 scores were consistently low across all three predictive tools evaluated.
- Given this pervasive low performance, it is highly probable that a larger sample size would simply reinforce the finding that these tools currently have limited predictive utility for AMR, rather than suddenly reveal a high-performing tool.
In conclusion, our study prioritized a deep, high-contrast evaluation across 780 distinct test cases over a wide, general population survey, and the resulting low-performance metrics are robust enough to draw strong conclusions regarding the limitations of the current marker gene-based methods for AMR prediction.
Comments 3:
The tables are very large and because of this it is difficult to perceive the information objectively - they need to be optimized or combined.
Reponse 3:
We appreciate the reviewer's feedback regarding the size and readability of the tables. We agree that clear data presentation is essential for the objective perception of information.
We prepared the tables to ensure that the detailed, isolate-specific comparisons required for the evaluation of AMR prediction tools were fully presented. The integrity of our study relies on the ability to trace the performance of each tool (Tables 3, 4, and 5) against the ground-truth characteristics (Tables 1 and 2).
- Tables 1 and 2 (Phenotype and Genotype Characteristics): These tables provide the essential, ground-truth data for the 20 E. coli isolates (the 780 test cases discussed in our previous response). These are the foundation for the entire evaluation and must remain distinct and highly detailed.
- Tables 3, 4, and 5 (Individual Tool Results): These tables are large because they present the specific performance metrics (e.g., predicted AMR status) for each of the 20 isolates for each of the three tools. Combining them would result in one massive, unreadable table.
- Table 6 (Comparative Features): This is a small, summary table of the tools themselves and is concise.
To address the reviewer's request for optimization while maintaining the necessary scientific detail, we propose moving the detailed, full results of the individual tool predictions (Tables 3, 4, and 5) to a Supplementary Appendix. This approach will streamline the main text for easier reading while ensuring that all the necessary raw data remains accessible for detailed scrutiny by interested researchers
Comments 4:
Also, you have a very high level of similarity (40%), it should be a maximum of 25-30%.
Reponse 4:
We will obtain the full similarity report (or originality report) from the journal editorial office. And then we will carefully review the full originality report, extensively paraphrased the flagged sections to ensure original wording, and confirmed that the similarity index is significantly reduced to an acceptable level (below 25-30%).
Comments 5:
Some comments in the attached file.
Response 5:
We have made revisions based on the review comments you provided, and furthermore will use the MDPI English editing service.
- Line 118: Ten carbapenem-resistant Escherichia coli (CRE) isolates and ten carbapenem-susceptible E. coli isolates were obtained for this study.
→ "Ten carbapenem-resistant Escherichia coli (CRE) isolates and ten carbapenem-susceptible E. coli isolates were obtained for this study, isolated and identified from patients hospitalized at general hospitals in Incheon, South Korea." - Line 122: In addition to carbapenem resistance, the VITEK® 2 system was used to determine the antimicrobial susceptibility profiles for a panel of other clinically relevant antibiotics.
→ "In addition to carbapenem resistance, the VITEK® 2 system (AST-N415) was used to determine the antimicrobial susceptibility profiles for a panel of other clinically relevant antibiotics (amoxicillin/clavulanic acid (CA), ampicillin, aztreonam, cefazolin, cefepime, cefotaxime, cefoxitin, ceftazidime, ciprofloxacin, ertapenem, imipenem, piperacillin/tazobactam, amikacin, gentamicin, trimethoprim/sulfamethoxazole, and tigecycline)." - Line 126: Genomic DNA was extracted from the bacterial colonies.
→ "Genomic DNA was extracted from the bacterial colonies were cultured from a suspension (1 g of feces mixed with 10 mL of PBS) on sheep blood agar at 37℃ for 24 hours under aerobic conditions, and on Gut microbiota media (GMM) and Gifu anaerobic medium agar, modified (GAM) at 35℃ for 48 hours in an anaerobic chamber (5% CO2​)." - Line 133: Two types of sequencing were performed on the extracted DNA.
→ "16S rRNA Gene Sequencing [4, 15] and Whole-Genome Sequencing (WGS) [11, 12, 13, 14] were performed on the extracted DNA." - Abbreviations and gene nomenclature:
→ We have also revised the abbreviations and gene nomenclature as suggested by the reviewer.
Round 2
Reviewer 1 Report
Comments and Suggestions for Authors
Authors have addressed most issues indicated in the initial review. Authors detailed issues with some of the major concerns and considering the article is for evaluation of the current tool available their rebuttal is acceptable.
Author Response
Comments 1:
Authors have addressed most issues indicated in the initial review. Authors detailed issues with some of the major concerns and considering the article is for evaluation of the current tool available their rebuttal is acceptable.
Response 1:
We sincerely appreciate your insightful comments and consideration.
Reviewer 2 Report
Comments and Suggestions for Authors
Dear colleagues,
It should be noted that the authors have revised the material well, but the tables are still very large and difficult to understand. I also ask you to make some corrections in the text, namely regarding the use of abbreviations - they should be used in the text after the first mention:
- The title of the article should not contain abbreviations - “Evaluation of marker gene-based in silico antimicrobial resistance (AMR) prediction tools”
- 38 - “Antimicrobial resistance (AMR); Marker gene-based prediction tools; Antimicrobial genes (ARGs); Escherichia coli; PICRUSt2; Tax4Fun; MicFunPred”
- 110 - “....within Escherichia coli Carbapenem-Resistant Enterobacteriaceae (CRE) colonies....”
- 119 - “...Ten CRE carbapenem-resistant Escherichia coli () isolates and ten...”
- 185 - “...from both 16S rRNA gene and Whole Genome Shotgun (WGS) sequencing were...”
- 198 - “...and prediction of antimicrobial resistance (AMR) genes were performed...”
...and so on throughout the text.
I also ask you to pay attention to minor typos in the text, such as extra punctuation marks or missing letters in words.
Sincerely,
Reviewer
Author Response
Comments 1:
Dear colleagues,
It should be noted that the authors have revised the material well, but the tables are still very large and difficult to understand. I also ask you to make some corrections in the text, namely regarding the use of abbreviations - they should be used in the text after the first mention:
- The title of the article should not contain abbreviations - “Evaluation of marker gene-based in silico antimicrobial resistance (AMR) prediction tools”
- 38 - “Antimicrobial resistance (AMR); Marker gene-based prediction tools; Antimicrobial genes (ARGs); Escherichia coli; PICRUSt2; Tax4Fun; MicFunPred”
- 110 - “....within Escherichia coli Carbapenem-Resistant Enterobacteriaceae (CRE) colonies....”
- 119 - “...Ten CRE carbapenem-resistant Escherichia coli () isolates and ten...”
- 185 - “...from both 16S rRNA gene and Whole Genome Shotgun (WGS) sequencing were...”
- 198 - “...and prediction of antimicrobial resistance (AMR) genes were performed...”
...and so on throughout the text.
I also ask you to pay attention to minor typos in the text, such as extra punctuation marks or missing letters in words.
Response 1:
We sincerely appreciate your insightful comments. The table size was downsized by narrowing the column spacing, and any parts that were difficult to understand were modified by adding explanations to the table titles. For other parts that needed correction, such as abbreviations or typos, we used MDPI's English proofreading service.